# Health Behaviors during the Early COVID-19 Containment Phase and Their Impact on Psychological Health

**DOI:** 10.3390/healthcare11142051

**Published:** 2023-07-17

**Authors:** Roni Elran-Barak, Dikla Segel-Karpas, Roi Estlein

**Affiliations:** Faculty of Social Welfare and Health Science, University of Haifa, Haifa 3498838, Israel; dsegel@univ.haifa.ac.il (D.S.-K.); restlein@univ.haifa.ac.il (R.E.)

**Keywords:** health behaviors, COVID-19, lockdown, psychological health

## Abstract

The COVID-19 lockdowns have brought significant changes to individuals’ daily lives, including their health behaviors and psychological health. Longitudinal studies exploring changes in health behaviors during the course of the initial containment phase are relatively scarce. Our aim is to understand how health behaviors have evolved during different phases of the early COVID-19 lockdowns and assess the impact of these changes on psychological well-being. By doing so, we hope to provide valuable insights that can enhance the understanding of the relationship between health behaviors and psychological health, with relevance not only to everyday life but to times of crises. A longitudinal study among 313 adults in Israel (44.5 ± 13.4 years old, 80% women) at three timepoints, beginning with the first COVID-19 lockdown (April 2020) and extending through June 2020. In each wave, participants were asked to report about exercising, eating fruits and vegetables, sharing family meals, and screen time. The BSI (Brief Symptom Inventory) was used to assess psychological health. There was an initial increase in the frequency of exercising (3.06 + 2.3 times a week) and shared meals (breakfast, 3.97 + 2.3; lunch, 5.30 + 1.9; dinner, 5.75 + 1.7 times a day) followed by a subsequent significant decrease in these behaviors (exercising, 2.84 + 2.0; breakfast, 2.63 + 2.1; lunch, 3.48 + 2.3; dinner, 4.75 + 2.0). The health behaviors of more exercising (*r* = −0.145, *p* = 0.43) and less screen time (*r* = 0.183, *p* = 0.010) had a positive impact on psychological health. External events, such as the first COVID-19 lockdown, may influence health behaviors which may, in turn, influence psychological health. While prior studies have mainly highlighted the negative impact of the pandemic on health behaviors, our analyses suggest that the first containment phase may have had an initial beneficial impact on several health behaviors, including exercising and family meals. However, this change was not sustainable.

## 1. Background

Health behaviors play a vital role in maintaining physical and mental well-being [1]. Numerous studies consistently suggest that engaging in positive health behaviors, such as regular exercise, healthy eating, and limited screen time [2,3], is associated with better physical and psychological health outcomes [4]. The Coronavirus Disease 2019 (COVID-19) lockdowns have brought significant changes to individuals’ daily lives, including alterations in routines, social interactions, and work environments [5,6,7,8,9]. These changes have also affected individuals’ health behaviors and psychological well-being [10]. However, detailed longitudinal data about changes in health behaviors during the early containment phase and the link between these changes and psychological health is scarce [11]. Given the important need to understand behavioral alterations and their impact on mental health during times of global crisis [5], the purpose of this study is to assess changes in health behaviors and their impact on different aspects of psychological health during the challenging period of the first COVID-19 lockdown.

The COVID-19 pandemic has introduced unique challenges that have influenced individuals’ health behaviors and psychological health [11]. The unprecedented nature of the pandemic, coupled with factors such as fear of infection, social isolation, disruptions in daily routines, and economic uncertainty, has contributed to an adverse impact on psychological well-being [10,12,13]. At the same time, measures such as social distancing, quarantine, and lockdowns have impacted people’s ability to engage in their usual health behaviors [5]. Additionally, heightened anxiety, fear, and uncertainty during the pandemic may have influenced individuals’ motivation and adherence to healthy behaviors, and vice versa—lack of health behaviors may have elevated peoples’ psychological health [6,7], and specifically—symptoms such as depression, anxiety, and somatization. Despite the importance of understanding the interplay between health behaviors and psychological health during the COVID-19 pandemic, there is a lack of empirical data in this area, and specifically—a scarcity of longitudinal data about the early containment phase. This knowledge gap hinders our ability to develop targeted interventions and support strategies to promote psychological well-being during unprecedented times.

Several studies [14,15] have specifically investigated the direct impact of COVID-19 lockdowns on health behaviors [16], suggesting that for some individuals, these lockdown measures have led to changes in health behavior [5]. In terms of physical activity [17,18,19,20], studies suggest that factors such as restrictions on movement, limited outdoor access, and gym closures during lockdowns could have contributed to a reduction in physical activity. But on the other hand, increased free time resulting from lockdown measures could have provided individuals with more opportunities to engage in exercising (e.g., extra free time, available online fitness programs). Several studies have confirmed a decrease in exercising during the COVID-19 lockdowns. A scoping review on physical activity and COVID-19 [21] has reported a decrease in physical activity levels due to social distancing measures, and a systematic review and meta-analysis of 22 international studies [20] confirmed these findings and revealed a decrease of 17 min per day in children’s moderate-to-vigorous physical activity from pre-pandemic to during the COVID-19 pandemic. However, some potentially contradicting findings exist. A study among students at a French university shows a favorable change in physical activity in about 20% to 25% of students and unfavorable changes in about 20% to 40% of students [7]. These potentially contradictory findings may arise from the broad examination of the pandemic as a whole without a detailed focus on the specific changes occurring during the initial containment phase. For instance, the rapid transition to working from home and the strict social isolation measures implemented during the initial containment phase of the pandemic may have resulted in individuals having more free time available, which in turn may have influenced their exercising routine differently than during subsequent phases of the pandemic. As promoting physical activity is a significant public health priority [19], it is crucial to gather more data on changes in exercise routines occurring during specific timepoints of global crises. This data will enhance our understanding and inform the development of health promotion interventions that promote physical activity both in everyday life and during times of crisis.

Additionally, prior studies have suggested that increased time spent at home during lockdowns has been associated with a rise in unhealthy food choices [16], a change in eating habits [22], as well as an increase in screen time [19]. A web-based survey among more than 10,000 French participants during the early COVID-19 containment phase suggests an increase of approximately 30% in caloric/salty food intake and an increase of about 65% in screen use [23]. However, an encouraging observation is that many households have reported an increased frequency of family meals during the lockdown periods [24]. With more family members staying at home due to remote work, school closures, or reduced social activities, there has been a notable rise in opportunities for shared meals. Results of a survey among about 700 adults suggest that maintaining regular family meals or increasing the frequency of these meals during the lockdowns was associated with increased closeness and more positive perceptions of the impact of the pandemic [24]. Therefore, we seek to identify changes in health behaviors during the early COVID-19 containment phase, understand the relationships between these changes and psychological well-being, and inform recommendations for health promotion interventions.

### 1.1. Research Questions

Based on this literature review, the present study aimed to address the following objectives among Israeli adults during the early COVID-19 containment phase:To describe changes in health behaviors, including exercising levels, dietary habits, family meals, and screen time.To describe changes in psychological health, including symptoms of depression, anxiety, and somatization.To discover the impact of the assessed health behaviors on psychological health.

Addressing these research questions is essential for several reasons. Describing changes in health behaviors occurring during the early COVID-19 containment phase while identifying the specific health behaviors that have the most significant impact on psychological health can inform recommendations for health promotion interventions, as well as public health campaigns and interventions. In addition, these findings can contribute to the broader understanding of the complex interplay between health behaviors and psychological health in the context of a global crisis.

### 1.2. Overview of COVID-19 Outbreak and Lockdown Measures in Israel: Summary of Events

The current study has taken place during the early containment phase of the COVID-19 outbreak. This paragraph provides an overview of the COVID-19 outbreak and the associated lockdown measures implemented in Israel during that time. The World Health Organization (WHO) declared the SARS-CoV-2 (the virus that causes COVID-19) outbreak an international public health emergency on 30 January 2020 and a pandemic on 11 March 2020 [25]. In an effort to contain this outbreak, the Israeli government announced a number of restrictions aimed at reinforcing social distancing [26]. On 19 March 2020, Prime Minister Benjamin Netanyahu declared [27] that social distancing restrictions would be legally enforceable, and violators would be fined. Israelis were forbidden from leaving their homes unless absolutely necessary. Essential services—including grocery stores, pharmacies, and banks—remained open, but people were prohibited from venturing more than 100 m from their homes, aside from under certain circumstances (e.g., stocking up on food and medicine). Non-essential stores were required to close, and parks were to remain shut. People were required not to participate in any social gatherings and to limit face-to-face interactions with individuals outside the immediate household [28]. Reports from Israel suggest that compliance with governmental orders during the first-wave lockdown was quite high [29,30]. The initial lockdown measures were implemented, as explained, on 19 March 2020 and were gradually eased starting on 19 April 2020. The easing of restrictions continued until July 2020, at which point the restrictions were lifted again.

## 2. Methods

### 2.1. Procedure

This study involves secondary analyses of data collected during the COVID-19 pandemic among cohabiting couples to examine the impact of the lockdown on couples’ relationships, dynamics, and well-being [31,32]. The research project received university institutional review board approval. Participants were recruited using a snowball sampling (i.e., by sending potential participants a link to the online questionnaire and asking them to share the link with others who might be interested in participating) and by posting an invitation on online forums and social media outlets, inviting individuals to take part in an online questionnaire exploring associations between health behaviors and couple dynamics patterns in the context of coping with the COVID-19 pandemic. Participants were required to provide online consent by pressing “agree” after reading the consent form, and they were eligible if (a) they were at least 18 years of age, (b) they had a romantic partner, and (c) the partners were cohabiting during the lockdown.

We employed a longitudinal research design for this study, using a three-timepoint web-based survey between 1 April and 14 June, 2020. Data were collected in Israel, with the first two points of data collection conducted during a lockdown with severe restrictions. The third wave was conducted after many of the restrictions had been lifted. The first-wave data (T1) were collected between 1 April and 14 April, the second-wave data (T2) were collected between 15 April and 30 April 2020, and the third-wave data (T3) were collected between 7 June and 14 June 2020. At the baseline assessment (T1), participants were asked to retrospectively report their health behaviors prior to the onset of the COVID-19 pandemic. This retrospective report, defined as T0, aims to capture participants’ behaviors before the pandemic-related changes occurred, providing a reference point for comparison with the subsequent assessments during the pandemic. The authors did not have access to information that could identify individual participants during or after data collection.

### 2.2. Participants

A total of 313 individuals (251 women, 62 men) participated in this study (Table 1). All participants were cohabiting with a romantic partner. Across the three waves of this study, the response rate for each survey ranged from 44% to 100%. Participants ranged in age from 24 to 81 years. The average age of the sample is 44.48 years, with a standard deviation of 13.4. In total, 80.4% of the sample is identified as female, 90.2% of the sample has a high level of education, and 16.6% of the sample is unemployed. Given the high rate of higher education among the participants, it is possible to characterize this sample as having a high socio-economic status.

### 2.3. Measures

Health behaviors were measured in three timepoints. In addition, in the first wave of data collection, participants were asked to recall their past behaviors. This method was used in other studies examining the impact of COVID-19 on health behaviors [24]. As we were unable to collect true longitudinal data on these questions, given the unpredicted nature of COVID-19, these retrospective responses were used to understand the influence of COVID-19 on participants’ practices. While current attitudes may influence participants’ recollections of their past well-being and behaviors [33], this method is helpful in understanding how participants viewed the impact of COVID-19 at the time of the survey and how they perceived they had changed because of it [24].

***Exercising:*** In each wave, participants were asked the following question: “How many times per week did you participate in sports activity or exercise for 30 min or more?” [0 to 7 days a week]. In the first wave, participants were also asked the following question: “Right before the onset of the COVID-19 pandemic, how many times per week did you participate in sports activity or exercise for 30 min or more?”

***Screen time:*** In each wave, participants were asked the following question: “Approximately, how many hours per day did you spend watching screens (e.g., computer, television) for non-work purposes?” [0 to 24 h a day]. In the first wave, participants were also asked the following question: “Right before the onset of the COVID-19 pandemic, approximately, how many hours per day did you spend watching screens (e.g., computer, television) for non-work purposes”.

***Fruit/Vegetable consumption:*** In each wave, participants were asked the following 2-item question: “On average, during the last week, how many fresh fruits/vegetables did you eat each day?” [0 to 10 per day]. In the first wave, participants were also asked the following 2-item question: “Right before the onset of the COVID-19 pandemic, how many times per day on average did you eat fresh fruit/vegetable?”

***Frequency of family meals:*** In each wave, participants were asked the following 3-item question: “How often did you eat breakfast/lunch/dinner with at least one other family member?” [0 to 7 days a week]. In the first wave, participants were also asked the following 3-item question: “Right before the onset of the COVID-19 pandemic, how often did you used to eat breakfast/lunch/dinner with at least one other family member?”

***The Brief Symptom Inventory (BSI)*** is a self-report questionnaire commonly used to assess psychological distress [34]. The short version consists of 18 items that measure various dimensions of psychological health, including somatization, depression, and anxiety. Participants rate the severity of their symptoms over the past week on a Likert scale. The BSI provides scores for different subscales as well as a global severity index, which provides an overall measure of psychological distress. The questionnaire has been widely validated and has demonstrated good reliability and validity in assessing psychological symptoms in various populations. In the current study, the reliability ranged from Cronbach’s alpha = 0.885 (T1) to 0.835 (T2) and 0.893 (T3).

### 2.4. Statistical Analyses

Data were analyzed using SPSS-23. A one-way repeated measures ANOVA was conducted to evaluate the impact of time on health behaviors and BSI scores when measured at three different timepoints: T1, T2, and T3. Follow-up paired *t*-tests were fitted to examine differences in health behaviors and psychological health in T0, T1, T2, and T3. Partial correlation analyses were conducted to investigate the influence of health behaviors at T1 on psychological health in each wave while controlling for age. This statistical approach allowed for the examination of the unique association between health behaviors and psychological health while accounting for the potential confounding effect of age. The results display the adjusted associations between the variables of interest. Age was selected as a control variable because of its significant associations with the BSI subscales (Table 1).

## 3. Results

The results section presents the statistical analysis conducted to examine changes in health behaviors and psychological well-being (measured using BSI) at three different timepoints during the early containment phases of COVID-19 in Israel. We will first present the results of one-way repeated measures ANOVA employed to assess the effect of time on health behaviors and psychological well-being. We will then present the results of partial correlation analyses conducted to explore the relationships between health behaviors at T1 and psychological health in each wave, with age as a control variable.

### 3.1. Changes in Health Behaviors and Psychological Well-Being

***Exercising:*** The results of the repeated measures ANOVA (T1, T2, T3) indicated a significant time effect, Wilks’ lambda = 0.838, F(2,51) = 4.9, *p* = 0.011, partial eta-squared = 0.162. A follow-up comparison indicated that there was an increase in the frequency of exercising when the early COVID-19 containment phase started and a decrease thereafter. Table 2 demonstrates that from T0 to T1, there was an increase in the frequency of exercising. From T1 to T2, there was no change in exercising, and from T1 to T3, there was a decrease in the frequency of exercising.

***Screen time:*** The results of the repeated measures ANOVA (T1, T2, T3) indicated a significant time effect, Wilks’ lambda = 0.710, F(2,65) = 13.3, *p* < 0.001, partial eta-squared = 0.290. A follow-up comparison indicated that there was an increase in screen time when the early COVID-19 containment phase started and a decrease thereafter. Table 2 demonstrates that from T0 to T1, there was an increase in screen time and that from T1 to T2 and from T1 to T3, there was a decrease in screen time.

***Frequency of family meals:*** The results of the repeated measures ANOVA (T1, T2, T3) indicated a significant time effect for family breakfast, lunch, and dinner. Wilks’ lambda ranged from 0.602 (breakfast) to 0.527 (lunch) and 0.717 (dinner). A follow-up comparison indicated that there was an overall increase in the frequency of shared meals when the early COVID-19 containment phase started and a decrease thereafter. Table 2 demonstrates that from T0 to T1, there was an increase in the frequency of shared meals during breakfast, lunch, and dinner. From T1 to T2, there was a decrease in the frequency of shared meals during breakfast, and from T1 to T3, there was a decrease in the frequency of shared meals during breakfast, lunch, and dinner.

***Fruit/vegetable consumption:*** The results of the repeated measures ANOVA (T1, T2, T3) indicated a non-significant time effect, Wilks’ lambda = 0.998 (fruits), 0.980 (vegetables). Table 2 shows that, overall, the early COVID-19 containment phase did not have a significant impact on fruit/vegetable consumption, except for a slight increase in vegetable consumption from T1 to T3.

***Psychological Well-being—Brief Symptom Inventory (BSI):***Somatization: The results of the repeated measures ANOVA indicated a significant time effect, Wilks’ lambda = 0.866, F(2,70) = 5.4, *p* = 0.006, partial eta-squared = 0.134. A follow-up comparison indicated significant differences from T1 to T2. Depression: The results of the repeated measures ANOVA indicated a significant time effect, Wilks’ lambda = 0.84, F(2,70) = 6.7, *p* = 0.002, partial eta-squared = 0.134. A follow-up comparison indicated significant differences from T1 to T2 and from T1 to T3. Anxiety: The results of the repeated measures ANOVA indicated a significant time effect, Wilks’ lambda = 0.88, F(2,70) = 4.6, *p* = 0.013, partial eta-squared = 0.117. A follow-up comparison indicated significant differences from T1 to T2.

### 3.2. Associations between Health Behaviors and Psychological Well-Being

When examining the impact of health behaviors on psychological health (while adjusting for age, Table 3), the following significant findings emerged. The impact of health behaviors at T1 on psychological health at T1: The partial correlation analysis revealed a significant negative association between sports activity and somatization (*r* = −0.145, *p* = 0.043), and screen time and depression (*r* = 0.183, *p* = 0.010), after controlling for age. This suggests that higher levels of sports activity (T1) were associated with lower levels of somatization symptoms (T1) and that higher levels of screen time (T1) were associated with higher levels of depressive symptoms (T1). The impact of health behaviors at T1 on psychological health at T3: The partial correlation analysis revealed a significant negative association between sports activity and somatization (*r* = −0.248, *p* = 0.029) after controlling for age. This suggests that higher levels of sports activity (T1) were associated with lower levels of somatization symptoms (T3).

## 4. Discussion

This study aimed to assess changes in health behaviors and their impact on psychological health during the early COVID-19 containment phase. While prior studies have mainly focused on the impact of the pandemic as a whole on health behaviors, our findings concentrate on three timepoints during the early containment phase. The findings suggest an initial improvement in some health behaviors, including exercising and shared family meals, but these improvements were not sustained over time. We also found that the health behaviors with the most significant impact on psychological health were exercising and screen time. It is worth noting that our study’s findings may be influenced by the composition of our sample, which predominantly consisted of females of higher socio-economic status. It is possible that these individuals had fewer financial worries during the pandemic, potentially impacting their health behaviors and psychological well-being differently. The discussion will focus on the key findings related to changes in health behaviors, their associations with psychological health, and the implications of these findings to health promotion research.

As expected, we found that there were changes in health behaviors during the early COVID-19 containment phase. However, in contrast to prior studies suggesting a decrease in exercising [20], our findings suggest an initial increase in the weekly frequency of exercising. While prior studies have mainly highlighted the negative impact of the pandemic on health behaviors, including exercising [5,6,7], our analyses suggest that the first containment phase may have initially had a beneficial impact on exercising, but these changes were not sustainable for the most part. For example, a study conducted among Swedish adolescents [14] suggests no significant differences regarding exercising before and after the COVID-19 outbreak, and a meta-analysis suggests an overall decrease [20]. However, it is plausible that the specific timing of data collection during the outbreak (such as the duration, characteristics, and severity of the containment measures) may have influenced these findings. Given that our results suggest a pattern of an immediate positive change followed by a decline in some health behaviors, it is important to exercise caution when making generalizations about changes (or lack of changes) in health behaviors during the outbreak.

These findings about exercising during the early phase of the lockdown can be attributed to various factors explaining why exercising may have been prioritized. These factors could include individuals’ desire for stress relief, increased free time, and a heightened awareness of the importance of maintaining physical health during times of crisis. Importantly, there was an increased availability of free time at home as a result of limited social activities, restricted movement, and reduced out-of-home work commitments or commuting time. With fewer options for out-of-home entertainment, exercise may have become a more accessible and convenient option for staying active and engaged. Additionally, the closure of gyms and fitness centers during lockdowns prompted individuals to seek alternative ways to maintain their fitness levels, such as home-based workout routines, online exercise classes, fitness apps, or home gym equipment. It could be that at later phases of the containment, people may have become less motivated or experienced accumulated pandemic-related stress, which could have contributed to a decline in physical activity.

Similarly, our findings indicate that the frequency of family meals increased in the first wave of data collection and subsequently declined in the next two waves among cohabiting couples. This direction was also proposed by a prior report [24]. It is reasonable to assume that family meals increased during the lockdown due to limited options for dining out. In addition, the synchronization of schedules due to restricted movement and reduced work commitments, as well as school schedules for kids, allowed family members to gather and have meals together, which may not have been possible during busier pre-lockdown routines.

Findings also revealed associations between health behaviors and psychological health. Higher levels of sports activity were associated with lower levels of somatization symptoms. And higher levels of screen time were associated with higher levels of depressive symptoms at T1 and somatization symptoms at T2, indicating a potential negative impact of increased screen time on individuals’ mental health. These findings are in line with previous research highlighting the beneficial effects of exercising on psychological well-being [4] and the potential negative consequences of excessive screen time on mental health [35]. Furthermore, the findings highlight no associations between shared family meals among cohabiting couples and psychological well-being during the COVID-19 lockdown. Although we did not find significant associations, public health initiatives can encourage and support families in prioritizing and maintaining regular shared meals, especially during times of crisis, as prior studies about family meals suggest that this routine allows family members to spend quality time with one another shared while fostering a sense of togetherness and connection with one another.

This study has several strengths, including its longitudinal design and the inclusion of multiple waves of data collection during the early COVID-19 containment phase. However, there are also limitations that should be acknowledged. One of the major limitations of this study is the exclusive focus on a subset of health behaviors, including physical activity, dietary habits, family meals, and screen time. We recognize that other crucial health behaviors, such as sleep patterns, time management, social relationships, alcohol consumption, and tobacco use, were not explored in this research. Consequently, this study’s findings provide only a partial understanding of the overall impact of health behaviors on psychological well-being during the early COVID-19 containment phase. Relatedly, there are various ways in which individuals can engage in physical activity (e.g., household chores, gardening) beyond conventional exercise practices, and while fruits and vegetables are essential components of a healthy diet, they are not the only factors to consider when assessing the overall balance and healthfulness of one’s diet. Future research should aim to include a more comprehensive assessment of various health behaviors. The sample was not representative, and participants were mainly females and of higher socio-economic status with high access to screens. This study relied on self-report measures, which are subject to biases and limitations in recollection. In addition, the observed associations between health behaviors and psychological well-being were relatively modest, and the study sample consisted of cohabiting couples, which may limit the generalizability of the findings to other populations. It is important to note that the impacts of the COVID-19 lockdown on health behaviors can vary among individuals and populations. Factors such as socio-economic status, living conditions, access to resources, and pre-existing health conditions can influence the extent and nature of these effects. Additional research is needed to verify our findings using a representative sample. Further research could also examine other possible outcomes of health behaviors, such as physical health, and/or follow participants for a longer period of time.

The findings of this study have important implications for public health campaigns and interventions during the COVID-19 pandemic. The results suggest that the pandemic has brought about a rapid change in multiple aspects of daily life, which, in turn, has led to an initial change in health behaviors. However, it is important to note that these initial changes in health behaviors were not sustainable in the long term. It appears that many individuals were unable to maintain these changes over an extended period. Factors such as fatigue, loss of motivation, the gradual easing of restrictions, and the resumption of regular routines may have contributed to the decline in sustained health behavior modifications. The transient nature of these behavioral shifts underscores the need for ongoing support and intervention from public health efforts. It is crucial for public health initiatives to provide individuals with the necessary resources, guidance, and strategies to maintain engagement in health behaviors over an extended period. By offering continued support, public health efforts can help individuals sustain positive health behaviors beyond the initial phase of rapid change and promote long-term well-being.

## 5. Conclusions

This study suggests that the first COVID-19 lockdown was initially linked to an increase in levels of exercise and the frequency of shared meals among cohabiting couples. However, these improvements were not sustained over time. We also found that the health behaviors with the most significant impact on psychological health were exercising and screen time. The findings of this pilot study imply that programs designed to improve health behaviors should include a dedicated component dealing with situations of an expected crisis, as these situations can lead to various changes in health behaviors.

Furthermore, our study opens avenues for future research in this field. Researchers can explore the long-term effects of crisis-induced changes in health behaviors and assess their sustainability beyond the initial phase. Investigating the factors that influence individuals’ ability to maintain positive health behaviors during and after crises can provide valuable insights into developing targeted interventions and support strategies. The practical implications of our findings extend beyond the current pandemic. By anticipating and addressing the unique challenges that arise during crises, we can better support individuals in maintaining healthy lifestyles and psychological well-being.

## Figures and Tables

**Table 1 healthcare-11-02051-t001:** Characteristics of participants and associations with health behaviors and psychological health at T1.

			Health Behaviors							Brief Symptoms Inventory		
			Sport Activity	Eat Fruits	Eat Vegetables	Family Breakfast	Family Lunch	Family Dinner	Screen Time	Somatization	Depression	Anxiety
Age	44.48	*r=*	**0.180**	0.114	−0.067	−0.009	0.038	**−0.165**	−0.075	**−0.165**	**−0.252**	**−0.200**
*Average (std)*	(13.4)	*p=*	**0.006**	0.067	0.277	0.887	0.540	**0.007**	0.220	** *p =* ** **0.006**	** *p =* ** **0.000**	** *p =* ** **0.001**
Sex	80.4%	*r=*	−0.065	−0.080	0.006	0.033	−0.104	0.072	0.011	0.077	0.023	0.001
*% Females*		*p=*	0.320	0.201	0.922	0.602	0.092	0.246	0.859	*p =* 0.206	*p =* 0.710	*p =* 0.993
Education,	90.2%	*r=*	0.022	**−0.156**	−0.088	−0.068	−0.057	0.019	−0.116	0.089	0.015	0.095
*% high*		*p=*	0.746	**0.014**	0.159	0.291	0.360	0.769	0.063	*p =* 0.151	*p =* 0.805	*p =* 0.122
Work status	16.6%	*r=*	0.040	0.071	0.009	**0.134**	**0.206**	−0.026	0.057	−0.074	−0.043	−0.050
*% Unemployed*		*p=*	0.547	0.255	0.887	**0.034**	**0.001**	0.673	0.353	*p =* 0.220	*p =* 0.481	*p =* 0.412
BMI	25.3	*r=*	**−0.215**	0.010	−0.036	0.019	0.062	−0.116	0.101	0.075	0.116	0.090
*Average (std)*	(5.0)	*p=*	**0.001**	0.877	0.568	0.775	0.327	0.069	0.109	*p =* 0.236	*p =* 0.066	*p =* 0.151

Note: Significant values appear in bold. The first-wave data (T1) were collected between 1 April and 14 April 2020.

**Table 2 healthcare-11-02051-t002:** Changes in health behaviors and psychological health.

			T0			T1			T2			T3		
	Min	Max	N	Mean	Std	N	Mean	Std	N	Mean	Std	N	Mean	Std
**Health Behaviors**														
Sport Activity ^1^ (times/week)	0	7	248	2.74	1.8	234	3.06 ^a^	2.3	117	3.23 ^ab^	2.4	110	2.84 ^b^	2.0
Eat Fruits (fruits/day)	0	10	259	2.07	1.5	259	2.30 ^a^	1.8	117	2.18 ^a^	1.7	120	2.54 ^a^	1.8
Eat Vegetables (vegetables/day)	0	10	272	2.82	1.9	267	3.15 ^b^	2.0	112	3.21 ^ab^	1.9	116	3.34 ^a^	2.1
Family Breakfast (meals/week)	0	7	218	1.84	2.0	249	3.97 ^a^	2.3	109	3.50 ^b^	2.5	104	2.63 ^b^	2.1
Family Lunch (meals/week)	0	7	270	2.09	2.1	266	5.30 ^a^	1.9	117	5.20 ^a^	2.0	113	3.48 ^b^	2.3
Family Dinner (meals/week)	0	7	268	4.55	2.1	261	5.75 ^a^	1.7	116	5.43 ^a^	2.1	110	4.75 ^b^	2.0
Screen Time (hours/day)	0	24	265	2.83	2.2	268	4.97 ^a^	3.7	118	3.75 ^b^	3.2	118	3.15 ^b^	2.8
**Brief symptoms inventory**														
Somatization	6	30				276	7.54 ^a^	2.2	120	7.09 ^b^	1.8	123	7.39 ^ab^	2.1
Depression	6	30				275	9.62 ^a^	3.4	120	8.77 ^b^	2.6	123	9.03 ^b^	3.6
Anxiety	6	30				275	10.32 ^a^	3.5	120	9.20 ^b^	3.0	123	9.47 ^b^	3.6

The first-wave data (T1) were collected between 1 April and 14 April 2020, the second-wave data (T2) were collected between 15 April and 30 April 2020, and the third-wave data (T3) were collected between 7 June and 14 June 2020. T0 data were collected during the baseline assessment (T1), where participants were asked to retrospectively report their health behaviors prior to the onset of the COVID-19 pandemic. ^1^ Sports activity or exercise for 30 min or more. ^ab^ Differing superscripts indicate significant differences.

**Table 3 healthcare-11-02051-t003:** Associations between health behaviors at T1 and psychological health at T1 and T3, adjusted for age.

		Brief Symptoms Inventory (T1)			Brief Symptoms Inventory (T3)		
		Somatization	Depression	Anxiety	Somatization	Depression	Anxiety
Sport Activity	* r= *	** −0.145 **	−0.140	−0.076	** −0.248 **	−0.165	−0.140
	* p= *	** 0.043 **	0.052	0.292	** 0.029 **	0.148	0.223
Eat Fruits	* r= *	−0.101	0.027	−0.065	−0.065	0.068	0.090
	* p= *	0.158	0.703	0.368	0.574	0.552	0.435
Eat Vegetables	* r= *	−0.057	−0.014	−0.052	0.028	0.098	0.202
	* p= *	0.431	0.842	0.474	0.807	0.394	0.076
Family Breakfast	* r= *	−0.079	0.095	0.052	−0.143	0.205	0.049
	* p= *	0.271	0.188	0.470	0.213	0.072	0.672
Family Lunch	* r= *	−0.078	0.037	−0.006	−0.022	0.198	0.083
	* p= *	0.279	0.604	0.934	0.845	0.083	0.468
Family Dinner	* r= *	−0.031	−0.013	−0.042	−0.036	0.037	0.045
	* p= *	0.666	0.854	0.556	0.754	0.746	0.699
Screen Time	* r= *	0.087	** 0.183 **	0.066	0.123	0.024	0.111
	* p= *	0.225	** 0.010 **	0.356	0.284	0.838	0.334

Note: The first-wave data (T1) were collected between 1 April and 14 April 2020, and the third-wave data (T3) were collected between 7 June and 14 June 2020. Bold indicates significant values.

## Data Availability

Data is available upon request from R.E.-B.

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
