# Peer review of "Health Behaviors during the Early COVID-19 Containment Phase and Their Impact on Psychological Health"

_healthcare, 2023, doi:10.3390/healthcare11142051_

Round 1

Reviewer 1 Report

This manuscript is well written and easy to read and discusses health behaviors during the early COVID-19 containment phase and their impact on psychological health. The authors aim to understand how health behaviors have evolved during different phases of the early COVID-19 lockdowns and assess the impact of these changes on psychological well-being.

I consider “Introduction” section interesting, and the references are sufficient and up to date (perhaps the authors could update with data from 2023), however, why focus only on physical activity, dietary habits/family meals and screen time (research questions)? Aren't sleep, time management, social relationships, alcohol, and tobacco consumption, etc. also part of lifestyle and health behaviors?

More so, is the concept of physical activity what the authors want? Because in the Measures subtitle (line 175) they talk about sports activity or exercise (“Exercising”). A person can be extremely active and do neither.

In addition, while eating fruits and vegetables is indeed an essential aspect of a healthy diet, it is not the only important factor to consider. A well-rounded and balanced diet encompasses a variety of food groups to provide all the necessary nutrients for optimal health.

Regarding the “Methods” section, it is essential to have more information about the selection of these two participants, is there no risk of selection bias here? If the authors are selecting participants in online forums and social media, can they exclude for example participants with little screen time? Is this sample representative? 

Concerning data collection, why did it take place in two weeks in the first two phases and only one week in the third phase? At baseline, could there be bias that participants are reporting retrospectively, prior to the onset of the covid19 pandemic?

Would it make sense to know the satisfaction with the couple relationship? All participants may be cohabiting with their partner, but the quality of their relationship may influence their habits and even their physical and psychological well-being.

Many of the considerations may not be changed now, but could be discussed in the discussion.

The associations found in the results are weak, so that conclusions should be drawn with caution. What does this study really bring? Why is this novel?

Line 327 à use “it is” instead of “it’s”

Reviewer 2 Report

I would like to thank MDPI for the possibility to review this work.

This is not new work, as many articles have been published on the effects of COVID-19 confinement.

The study is significantly biased by the recruitment method used, and perhaps explains why 80% of the sample is female. The authors should work this issue into the discussion.

Measuring instruments are appropriate and used correctly.

A repeated measures analysis with repeated measures ANOVA is missing in the statistical analysis, as many of the variables are continuous and may follow a normal distribution. This study would allow to see within-group and within-group differences such as differentiating by sex.

Reviewer 3 Report

Thank you very much for give me the opportunity to review this interesting paper titled Health behaviors during the early COVID-19 containment  phase and their impact on psychological health.

Altogether it is an interesting study quite well written and with a complex procedure, using a three-time point web-based survey. It also presented a detailed analysis of the changes in health behaviors population in Israel, applying paired t-test to examine differences in health behaviors and psychological health in t1, t2, and t3 and conducting partial correlation analyses to investigate the influence of health behaviors at T1 on psychological health in each wave, while controlling for age. As authors noted Longitudinal studies exploring changes in health behaviors during the course of the initial containment phase are relatively scarce.

The objective is well defined, and citations are current.

The sample size is quite high, and the statistical analysis are coherent.

Say this, here are a few comments:

1.      The abstract contains important information and results from the study, but perhaps the incorporation in number of what appear to be averages and standard deviations, or of correlations (some of them is wrong, maybe it must be 0.043) in my humble opinion makes it difficult to read it. Perhaps it would be interesting to rewrite it without incorporating such data.

2.      Could you please review line 57 what does “scary” mean in such line? It you wanted to write “scarcity” maybe you could re-write the whole sentence, in example, “there is a scarcity of empirical data in this area, and specifically of longitudinal data about the early containment phase”

3.      In lines 69-77 you offer several results related to the objective of your study, but among participants that are exactly the one you are analyzing. It would be interesting if, in a similar vein to the studies indicated in lines 81-94, if possible, you could synthesize and describe more studies that are closer to your sample and the questions you are investigating in it. This could not only improve the introduction itself, but also the set of references consulted and even the conclusions section.

4.      In my humble opinion the introductory section should be improved, with more detail on other research and its conclusions, perhaps even relocating subheading 1.1. in a different way, I understand that you want to show the impact of the confinement measures in Israel on behaviors such as exercise, but as it is now the paragraph seems a bit disconnected from the rest of the text.

5.      In Table 1 you show descriptive data of the sample but also correlational results of your study at T1, without having yet offered the description of your measurements, perhaps it would be interesting if you would relocate at least those correlational data in the results section.

6.      In the description of your measures, you do not provide reliability results on your sample (e.g., Cronbach alpha). Some of these measures contain only one item, but for others, such as the BSI, it would be useful to provide such data in your own sample.

7.      In my humble opinion, it would be convenient to include a short introduction in the results section to help the reader. In a similar vein, perhaps it would be interesting instead of ordering the information first by type of analysis (e.g., descriptive and correlational, and analysis of change with t-tests).

8.      I'm a little confused with your description and analysis of measurement moments. Please explain more clearly, maybe already in the procedure section, and not only in measurements, what is time 0 in your study.

9.      Maybe the conclusion section might be improved, rearranging them a bit and adding some more arguments regarding the practical implications and suggestions for future research of your work.

Round 2

Reviewer 1 Report

As reviewer, I am pleased to accept the paper in its current form, as the authors have diligently incorporated and complied with all of my suggestions.

Author Response

Thank you for your positive feedback and for your careful consideration of our manuscript.

Reviewer 2 Report

  • The work can be accepted in its present state

Author Response

(The authors gave the same response as above.)

Reviewer 3 Report

Thank you again for giving me the opportunity to review this interesting article. In my opinion the authors have clearly improved the manuscript. Just one small thing, please check if the format of your tables follows the one indicated by the journal, and if the spacing is correct.

Author Response

Thank you for your positive feedback and for your careful consideration of our manuscript. Upon reviewing the tables and spacing, we have ensured that they adhere to the guidelines provided by the journal. However, we are open to making any necessary changes if required by the editorial team.